# Green-Graphene Protective Overlayer on Optical Microfibers: Prolongs the Device Lifetime

**DOI:** 10.3390/nano12172915

**Published:** 2022-08-24

**Authors:** Anastasia Novikova, Aviad Katiyi, Aviran Halstuch, Alina Karabchevsky

**Affiliations:** School of Electrical and Computer Engineering, Ben-Gurion University of the Negev, Beer-Sheva 8410501, Israel

**Keywords:** green graphene, microfiber, gold nanoparticles

## Abstract

Optical microfibers find new applications in various fields of industry, which in turn require wear resistance, environmental friendliness and ease of use. However, optical microfibers are fragile. Here we report a new method to prolong the microfiber lifetime by modifying its surface with green-extracted graphene overlayers. Graphene films were obtained by dispergation of shungite mineral samples in an aqueous medium. For this, we tapered optical fibers and sculptured them with graphene films mixed with gold nanoparticles. We observed that due to the surface modification the lifetime and survivability of the microfiber increased 5 times, as compared to the bare microfiber. The embedded gold nanoparticles can also be utilized for enhanced sensitivity and other applications.

## 1. Introduction

Optical fibers are dated back to 1880 when William Wheeler transmitted light through a glass pipe and referred to it as ’light piping’. It took until 1966 when the optical fibers with a higher guiding medium as compared to the cladding were proposed for light transmission [1]. This development was intended mainly for telecommunication due to low losses and ease in fabrication [2]. Furthermore, optical fibers can be tapered to microfiber dimensions for experiencing the novel properties [3]. Microfiber [4] can be used for a verity of applications such as sensing [5,6,7,8], determination of substances [9], human health monitoring [10] and many others. One of the applications in which microfibers are important is sensing. Due to the squeezing of the fiber diameter, the confinement of the mode decreases and the evanescent field penetration depth to the analyte increases which in turn improve the sensitvity [11,12]. This can be utilized for sensing with tapered fibers [11,13,14].

These days, optical microfibers are widely used for the determination of various substances with low concentrations. Namely, in biomedicine [15] and biology [16] (low concentrations of viruses [17,18], bacteria [19,20], proteins [21], nucleic acids [22], cancer cells [23], substances in body [24]), environmental protection [25] (pollutants in water and soil, components of biological pollution) [26,27,28,29,30], pharmacology and the chemical industry (pharmaceutical substances, new materials) [31,32], construction industry (concrete deformation/stress measurements [33], the use of fiber-optic sensors for detecting railway vehicles and monitoring the dynamic characteristics of the rock mass caused by railway rolling stock for the needs of civil construction [34,35], damage detection and characterization using long-gauge and distributed fiber optic sensors [36], a new tool for temperature measurements in boreholes [37,38]); in addition, optical sensors are used in the creation of smart fabrics [39], stratigraphy [40,41] (probe for rapid snow grain size, determination of layered, sedimentary and volcanogenic rocks).

With such wide uses of microfibers, there are also different methods for modifying them, depending on the required parameters. One of the simplest methods is the method of chemical modification when various materials are deposited on the microfiber surface [42]. The use of microfibers is effective, due to their high sensitivity to the materials under study, and environmentally safe since microfibers consist of SiO and can be easily disposed of. In addition to the above-mentioned advantages, when using optical microfibers, little time is needed to determine the substances under study. Previously, studies were carried out with various microfibers and the optimal diameter of high sensitive microfiber was calculated as a diameter of 2.5 μm [11]. However, there is a challenge related to the small diameter of the microfiber. First, researchers need to be very careful when working with such a fragile device. Second, when working with metal nanoparticles and highly toxic substances, the microfiber can also be easily destroyed during the research process. The integration of gold nanoparticles with microfiber can achieve higher sensitivity due to the resonant behavior of metallic nanoparticles [11]. While using this type of microfiber on a large scale, these disadvantages can be significant. A solution to this problem can be the chemical modification of the microfiber surface with various substances [43]. These are mainly variations from nanoparticles of metals; recently, graphene and graphene oxide have also found wide applications. Graphene is mainly used not as a protective layer, but as a sensitivity enhancer of microfiber [18,44,45].

Here, we report the use of graphene films with gold nanoparticles to increase the lifetime of microfiber as illustrated in Figure 1. This type of microfiber is environmentally friendly, safe, easy to use and affordable. We deposited graphene films on the surface of microfiber, increasing its lifetime, which prevents the rapid destruction of microfiber by gold nanoparticles. In addition, the graphene films do not interfere with the determination of the peaks of gold nanoparticles. The increase in the lifetime of microfiber makes it possible to use it on an industrial scale for the determination of chemicals and microbiological objects. Furthermore, the embedded gold nanoparticles on the graphene films can be used for enhancing the sensitivity of the microfiber.

## 2. Experimental Section

### 2.1. Fabrication of the Sensing Device

In our work, we used tapered fiber as our sensing device. A single-mode fiber (SMF-28) was tapered using a commercial Vytran GPX-3000 tapering system. In the first step, we removed the acrylate coating from the fiber section where the tapering was carried out. After the removal of the coating, the fiber narrowed from a diameter of 125 μm to a diameter of ∼15 μm. In the next step, the fiber was narrowed to a diameter of ∼2.5 μm. The fiber was then attached to a metal fork with epoxy glue to prevent the fiber from breaking during movement. To apply and hold the graphene samples with gold nanoparticles, we placed a Teflon spacer under the microfiber region.

### 2.2. Obtaining Composite Materials

For obtaining the composite material, 0.01 mL of nanoparticles of gold with a diameter of 30 nm was added to plastic eppendorf with a capacity of 5 mL and 4.5 mL of distilled water was added. Dispersion of nanoparticles was carried out using a Sony ultrasonic bath in a solution of distilled water. Plastic tubes with the obtained suspensions were placed in an ultrasonic bath. Having dispersed the nanoparticles, we broke up the microagglomerates, increasing the specific surface area of the nanoparticles. In flasks with a capacity of 50 mL, 25 mL of distilled water and 4.51 mL of dispersed gold nanoparticles were added. Then, 1 g of shungite powder was placed in each flask and placed in an ultrasonic bath for 1 h. The resulting solutions were stored at room temperature.

### 2.3. Application of a Composite Material to the Microfiber Surface

For applying the composite material on the microfiber region, we applied 6 micro liters of composite material (graphene films + gold nanoparticles) to the microfiber region of the tapered fiber with an automatic pipette. Then, the laser was coupled to the tapered fiber and turn on. Under the thermal action of the laser, the water from the composite evaporated and the particles settled on the surface of the microfiber. The process of physical adsorption on the surface of graphene films and addition to OH groups also took place.

## 3. Results

To increase the lifetime of the microfiber while in contact with metals and caustic materials, we used graphene films, which prevent the microfiber from contacting with gold but do not change the peaks of gold nanoparticles during research. We obtained graphene films from the natural mineral shungite by the method of sonication in an aqueous solution without the addition of surfactants. We studied the surface of shungite samples before and after dispergation treatment to determine how the surfaces of the samples changed by scanning and transmission electron microscopy.

Figure 2 shows scanning and transmission electron microscopy images of shungite particles before and after the dispergation. Figure 2a shows a scanning electron microscopy image of the shungite sample (98% of carbon in amorphous form) before dispergation. It shows that shungite is homogeneous with small mineral inclusions. Figure 2b shows a transmission electron microscopy image of the shungite sample after dispergation. It shows shungite samples were stratified into thin films, and the specific surface area of the particles increased. These results are compatible with our previous work that showed that the shungite sample after dispergation contains graphene films and graphite-shaped parts [46]. These data will be confirmed later with Raman spectra and XPS data.

For the chemical modification of the fiber surface, we use graphene films with gold nanoparticles coating. We added gold nanoparticles with a diameter of 30 nm to the graphene samples obtained during sonication. The concentration ratio of graphene and gold was 1:1. Gold particles are adsorbed on the surface of graphene layers by physical sorption and chemical reaction due to the replacement of OH groups in graphene layers by gold. To investigate the interaction between the graphene films and the gold nanoparticles, we studied samples of graphene with gold nanoparticles and investigate them using transmission electron microscopy (TEM).

Figure 3a shows a transmission electron microscopy image of gold nanoparticles deposited on the graphene surface with a concentration of 1:1. It shows that the concentration is too low for developed graphene surfaces. As a result, we increased the ratio of graphene and gold nanoparticles to 1:3. Figure 3b shows a TEM image when the ratio of graphene films and gold nanoparticles in the sample is increased to 1:3. It shows that gold nanoparticles are deposited on the surface. Zoom image Figure 3c shows that the gold nanoparticles are embedded in graphene films and not separated from the graphene films. Therefore, we concluded that a ratio of 1:3 is good for surface modification. Figure 3d shows microfiber with layers of graphene and gold nanoparticles deposited on the surface. It shows that the graphene layers are unevenly distributed over the surface in separate spots.

Next, we study the samples of graphene with gold nanoparticles via UV-VIS spectrometry and Raman spectroscopy to determine the spectra of graphene and gold and how they affect each other in a composite material.

Figure 4a shows the transmission UV-VIS spectra of graphene and graphene with gold nanoparticles with a diameter of 30 nm. The absorption dip of the gold nanoparticles appears at 518 nm as compared to graphene without gold nanoparticles. The absorption dip of graphene tends towards 270 nm [47] and cannot be seen in UV-VIS spectroscopy. Therefore, in the next step, we investigated the sample using Raman spectroscopy. Figure 4b shows Raman spectra of the sample excited at 532 nm. D1=1345.56 cm${}^{-1}$ (graphite form, deformed form), G1=1638.36 cm${}^{-1}$ (graphene form, sp2) peaks refer to graphene films doped with graphite form. The intensity of the graphene form is higher as compared to the graphite form; accordingly, the presence of graphite is insignificant. The peak of gold nanoparticles does not observe since gold does not have peaks in the Raman spectrum. However, we can see a decrease in the D peak, since OH groups, which were presented in graphene, were replaced by the gold nanoparticles. The other peaks are related to water and the cuvette in which the sample was tested. Figure 4c shows the spectrum samples of graphene before the deposition of gold nanoparticles. The peaks are located in the same places—D1=1345.56 cm${}^{-1}$, G1=1638.36 cm${}^{-1}$—but the intensity of the defect layers (D-peak) is higher, and the graphene peak (G-peak) is lower. It confirms that gold nanoparticles were attached to graphene, partially replacing the OH group. These data are also confirmed by x-ray spectroscopy.

Next, we studied the samples using X-ray photoelectron spectrometry to determine the composition of the samples, characterize the carbon, and check how the oxygen concentration changes after the addition of gold nanoparticles. The data are presented in Table 1 and Table 2.

Comparing the data given in Table 1 and Table 2, we see that the gold nanoparticles peak appeared at 89.22 eV, Area 630.82 CPS·eV. The carbon peaks at 284.74 eV and 284.45 eV refer to the graphene peak [48,49]. The samples also contain small admixtures of titanium and oxygen, since the dispergation process took place in water. The full width at half maximum of the spectral line (FWHM) was 1.62 eV after the addition of gold nanoparticles changed to 1.46 eV. The percentage of gold in the total composition was 0.12 percent. In addition, when gold nanoparticles were added, the oxygen concentration decreased by 2.52 percent, which confirms that the addition of gold occurred in OH groups.

Here, we used the microfiber region of a tapered fiber as the sensing device. A commercial single-mode fiber was tapered to a diameter of 2.5 μm with a length of 2 mm. The fiber was glued to a metal fork for robustness and stability. To investigate different samples with different ratios of gold nanoparticles to graphene, we built the experimental setup shown in Figure 5a. A broadband laser was coupled to a single mode fiber via ×10 objective and was aligned for maximal power. The fiber was spliced with the tapered fiber sensing device which is made from a commercial single-mode fiber that was tapered to a microfiber with a diameter of 2.5 μm. The microfiber region acts as the sensing region for samples characterization due to the high evanescent field in this region. The output of the tapered fiber was spliced to a pigtail single-mode fiber that was connected to an optical spectrum analyzer. Figure 5b shows the tapered fiber sensing device coupled to the laser while the bright spot in the center is the microfiber region.

For verifying the optimal concentration of gold nanoparticles, we measured graphene with gold nanoparticles with different concentrations using a cuvette. We measured samples with ratios of graphene to gold nanoparticles of 1:1, 1:2, 1:3, 1:4, 1:5 and 1:6. Figure 5c shows VIS-NIR measurements of graphene and graphene with gold nanoparticles with different ratios of 1:1, 1:2, 1:3, 1:4, 1:5 and 1:6. While focusing on the wavelength range of 1300–1700 nm (Figure 5d), we can see the effect of different concentrations of gold nanoparticles on the graphene peak. The peak of graphene is shown at 1680 nm. At a ratio of 1:1 graphene to gold, the intensity decreased and the peak is shifted to the right, which indicates that the concentration of gold nanoparticles is too low. At a ratio of 1:2 and 1:3 with an increase in the concentration of gold, the intensity also increased and the peaks are also shifted to the right. For 1:4, 1:5 and 1:6 concentration, the absorption is slightly increased but with a large amount of gold nanoparticles. Therefore, the optimal ratio for increasing the intensity of the peaks is 1:3, since the peak is quite well pronounced and a large amount of gold nanoparticles is not used. Therefore, the optimal ratio to increase the absorption peaks is 1:3. After finding the optimal concentration of gold nanoparticles in graphene, we measured the transmission of the sample using a tapered fiber. A volume of 6 μL was dripped on a Teflon spacer and the microfiber was immersed in that volume. The VIS-NIR transmission was investigated for two samples (sample 1 and sample 2) with concentrations of 1:3. Figure 5e shows two graphs measurements of graphene with gold with concentration of 1:3 graphene to gold. The absorption peak at 560 nm is associated with the surface plasmon excitation of gold nanoparticles with a diameter of 30 nm. The peak is clearly visible as in the first and second sample. This means that with this ratio of graphene to gold, graphene does not interfere with the measurement of gold peaks.

Since it is known that microfibers are fast to break when interacting with metals and caustic substances in water [50,51], we investigated their lifetime with gold nanoparticles deposited on them while stabilized with graphene films. We applied 6 micro liters of different concentrations of gold nanoparticles on the microfiber region of the tapered fiber. We calculated the lifetime by measuring the transmission of the tapered fiber until the microfiber completely ruptured which caused the transmission to become zero. With a series of samples of graphene composites and gold nanoparticles, we performed the same procedure and measured the operating time of the microfiber until destruction. When applying gold nanoparticles with a diameter of 30 nm, the lifetime of the microfiber is 7 s. When graphene and gold nanoparticles were applied to microfiber, the microfiber lifetime was increased to 32 s. The experiment was repeated with concentrations of graphene to gold 1:1 1:2, 1:3, 1:4 1:5 and 1:6 in several repetitions. Regardless of the concentrations of gold, the lifetime of microfiber was the same, most likely since the concentration of graphene films in all samples was the same. From the data obtained, we conclude that the presence of graphene films on the surface of microfiber increases its lifetime times 4.57. Further optimization of the concentration may lead to an even longer lifetime of the fiber.

## 4. Discussion

We examined the use of graphene films from the natural mineral shungite with the addition of gold nanoparticles of gold to increase the lifetime of microfiber (SMF-28). From the data obtained, we can conclude that the presence of graphene particles on the surface of microfiber increases its lifetime by 4.57 times. This method opens up the prospect of utilizing microfibers with graphene for sensing and for the studies of corrosive substances and metals.

We conducted a study of graphene composites and selected the optimal ratio of gold nanoparticles to graphene (gold concentration: 600 μL—1:3; graphene concentration: 200 μL) at which the peaks of gold are clearly visible, but the microfiber does not degrade quickly which gives time for a more detailed study of the peaks before the destruction of microfiber takes place.

This type of microfiber with composites applied to it can be used for research in various industries, such as environmental protection (determination of various pollutants in water, soil and air), biotechnology and biology (determination of nucleotide sequences, proteins, viruses, bacteria), chemical industry (determination of new substances with low concentrations), pharmacology (study of new drugs with low concentrations), chemical industry (study new materials), construction industry (measurements of deformation and stress of materials, detection and characterization of damage, a new tool for measuring temperature and pressure), stratigraphy (probe for rapid snow grain size, determination of layered, sedimentary and volcanogenic rocks) and manufacturing of smart materials; it can also be used to capture sound waves.

## Figures and Tables

**Figure 1 nanomaterials-12-02915-f001:**
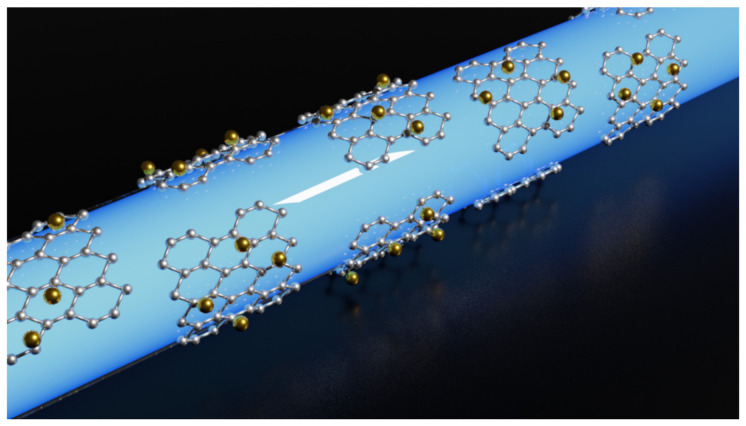
Artistic impression of microfiber with a protective overlayer of a composite of graphene and gold nanoparticles.

**Figure 2 nanomaterials-12-02915-f002:**
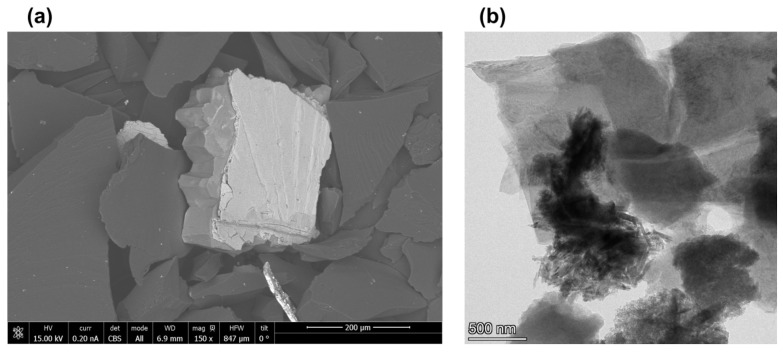
(**a**) Scanning electron microscopy (SEM) image of shungite particles before dispergation treatment and (**b**) transmission electron microscopy (TEM) image of graphene layers obtained from shungite.

**Figure 3 nanomaterials-12-02915-f003:**
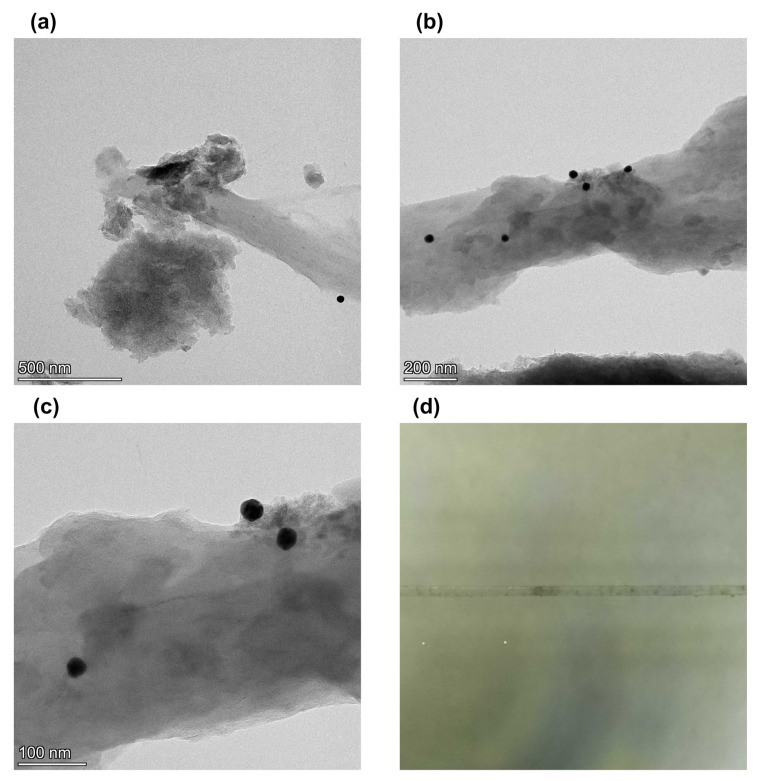
Transmission electron microscopy images of (**a**) graphene from shungite particles with gold nanoparticles with gold to graphene ratio 1:1, (**b**) graphene form shungite particles with gold nanoparticles with 3:1 ratio of gold to graphene and (**c**) zoomed image of graphene form shungite particles with gold nanoparticles with ratio of gold to graphene 3:1, (**d**) photo of microfiber with graphene and gold nanoparticles deposited on the surface.

**Figure 4 nanomaterials-12-02915-f004:**
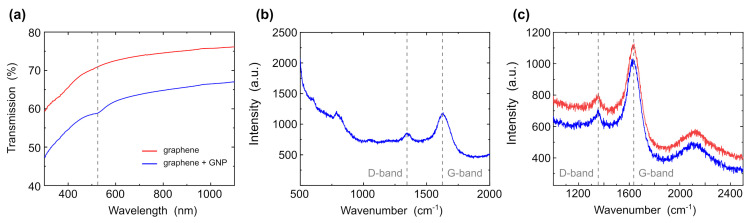
(**a**) UV-VIS spectra of graphene (red curve) and graphene with gold nanoparticles (blue curve). (**b**) Raman spectra of graphene with gold nanoparticles. (**c**) Raman spectra of graphene of two different samples.

**Figure 5 nanomaterials-12-02915-f005:**
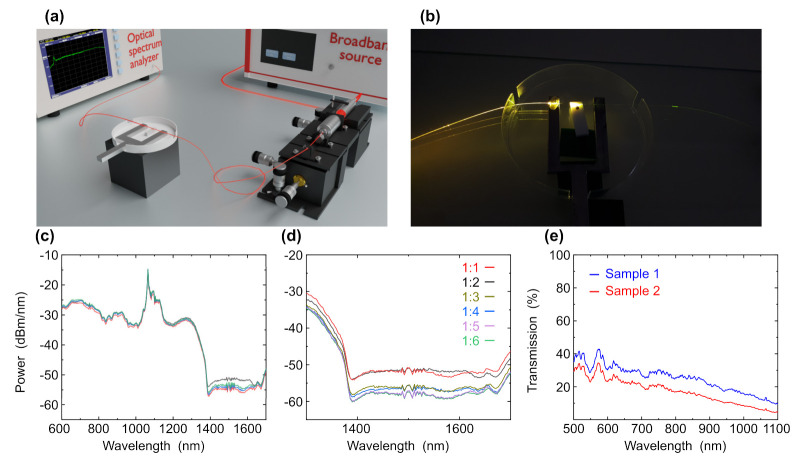
(**a**) Experimental setup used in the tapered fiber experiment. (**b**) The tapered fiber structure with the sample on the Teflon spacer. (**c**) VIS-NIR spectra of graphene and graphene with gold nanoparticles in cuvette at wavelength range of 600–1700 nm and (**d**) zoom-in intersection region at wavelength range of 1300–1700 nm with ratio of graphene to gold nanoparticles of 1:1, 1:2, 1:3, 1:4, 1:5 and 1:6. (**e**) Spectra of graphene overlayers on tapered fibers based on graphene form of shungite and gold nanoparticles at wavelength range of 500–1100 nm.

**Table 1 nanomaterials-12-02915-t001:** Elements composition and quantitative evaluation of samples of treated shungite.

Name	Peak (BE)	FWHM (eV)	Area (CPS·eV)	Atomic (%)
C1S	284.74	1.62	16,460.49	56.92
O1S	532.11	2.33	32,044.85	40.95
Ti2P3	458.70	1.16	6125.11	2.13

**Table 2 nanomaterials-12-02915-t002:** Elements composition and quantitative evaluation of samples of treated shungite with gold nanoparticle.

Name	Peak (BE)	FWHM (eV)	Area (CPS·eV)	Atomic (%)
C1S	284.45	1.46	16,150.69	56.34
O1S	531.70	3.28	34,194.57	38.43
Au4F	89.22	6.78	630.82	0.12
Ti2P3	458.40	1.24	7768.65	5.10

## Data Availability

Data underlying the results presented in this paper are not publicly available at this time but may be obtained from the authors upon reasonable request.

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
