# Peer review of "Green-Graphene Protective Overlayer on Optical Microfibers: Prolongs the Device Lifetime"

_nanomaterials, 2022, doi:10.3390/nano12172915_

Round 1
Reviewer 1 Report
The manuscript reports to prolong the microfiber lifetime by modifying its surface with green-extracted graphene overlayers. In its current version, there are many issues need to addressed:
1. English writing should be significant improved.
2. There are many repeated contents. Just list an example here, words in lines from 61 to 67 are the same with that in lines 155-161. Please check though the whole manuscript.
3. How do you conclude that graphene and small portion of mineral was obtained from the SEM and TEM images? I think you can only make this conclusion from Raman spectrum and EDS.
4. The lifetime of the microfiber is increased from 7s to 32s when graphene and gold nanoparticles were applied to microfiber. Although the life time is increased, the fiber is broken anyway, what is the application of this technology? The authors should also explain why the microfiber is broken when it is contracted with metals.
5. The diameter of the microfiber should have significant influence on its life time, which has to be investigated. The two samples coated with gold nanoparticles only and composite material of graphene and gold nanoparticles may have different tapered diameter, hence repeatability test is also required to draw the conclusion.
Reviewer 2 Report
It is meaningful to prolong the lifetime of optical microfibers. This work reported a green method of extracting graphene, and applied the green-graphene to microfibers. Based on the properties of graphene, the authors used the graphene film as a protective layer. It can not only protect the microfiber, but also prevent the rapid destruction of microfiber by the gold nanoparticles. Owing to the graphene overlayer, the device life of optical microfiber is increased five times. This work offers a new route to use microfibers with graphene for sensing and metals corrosive research. However, several issues should be addressed
Major concerns:
1. What is the standard for microfiber lifetime? How to evaluate and compare the liftime of microfiber? Please describe more in the manuscript.
2. It is very nice to use Raman to confirm the existence of graphene in figure 4. However, due to interference with other carbon materials, that is not enough as evidence. In order to confirm that the main component of the sample is graphene, the author should further test XRD and so on.
3. Line 109, line 136, line 142 and line 154. The author mentioned several times “the replacement of OH groups in graphene layers by gold”. In this way, gold and graphene will form new chemical bonds. XPS will show the corresponding new peak. Please use figures instead of tables, and mark the position of the peak in XPS figures.
Minor concerns
1. Is the thermal action of the laser only the evaporation of water? If there are other functions, please add them to the manuscript
2. Please unify and correct the form of references.
Reviewer 3 Report
The paper entitled "Green-Graphene protective overlayer on optical microfibers: prolongs the device lifetime" by Novikova et al. presents a method to improve the optical microfibers lifetime. The topic can be of interest, but the overall methodology and results are poorly presented and the conclusions are not supported by the presented results. Moreover the text presents several typos and the method section is repeated in the results section (from row 84 to row 95, page 3 was already written in the 2.2 obtaining composite materials subsection, and from row 155 to row 161 was already written in the 2.1 subsection)
Apart from this, the authors do not provide a sounding explanation about why the ratio graphene/ gold nanoparticles of 1:3 is the best option: in which terms? It represent the best option in term of quality layer, or in term of increasing the fiber lifetime? For example, a comparison with the higher gold contents is not discussed.
Moreover, it is missing a clear label of data reported in figures 4 and 5: for example, in fig 4 c which sample is represented by the blue line and which by the red line? In fig 5c what are sample 1 and sample 2?
Finally, the claim at page 7, row 193: “Since it is known that this microfibers fast to brake when in contact with metals and caustic substances in water[49,50], we investigated their lifetime with gold nanoparticles deposited on them while stabilised graphene films. When applying gold nanoparticles with a diameter of 30 nm, the lifetime of the microfiber is 7 seconds. When graphene and gold nanoparticles were applied to microfiber, the microfiber lifetime was increased to 32 seconds. The experiment was repeated with concentrations of graphene to gold 1:1 1:2, 1:3, 1:4 1:5 and 1:6 in several repetitions. From the data obtained, we conclude that the presence of graphene films on the surface of microfiber increases its lifetime times 4.57.” is not supported by any experimental figures or data, nor a description of the experimental method to measure the microfiber lifetime.
Round 2
Reviewer 1 Report
My comments are well addresed